# Genetic diversity and population structure of Uganda's yam (*Dioscorea spp*.) genetic resource based on DArTseq

**Emmanuel Amponsah Adjei**[1,2,3]*, **Williams Esuma**[4], **Titus Alicai**[4‡],
**Ranjana Bhattacharjee**[5‡], **Isaac Onziga Dramadri**[1,2‡], **Richard Edema**[1,2‡], **Emmanuel Boache Chamba**[3‡], **Thomas Lapaka Odong**[1]

**1** Department of Agricultural Production, Makerere University, Kampala, Uganda, **2** Makerere University Regional Center for Crop Improvement, Makerere University, Kampala, Uganda, **3** Council for Scientific and Industrial Research—Savanna Agricultural Research Institute, Kumasi, Ghana, **4** National Crops Resources Research Institute, Kampala, Uganda, **5** International Institute of Tropical Agriculture, Ibadan, Nigeria

☯ These authors contributed equally to this work.
‡ These authors also contributed equally to this work
* emmaadjei1@gmail.com

**Data Availability Statement:** All data files are available from the yambase database (https://yambase.org/).

## Abstract

Assessing the genetic diversity of yam germplasm from different geographical origins for cultivation and breeding purposes is an essential step for crop genetic resource conservation and genetic improvement, especially where the crop faces minimal attention. This study aimed to classify the population structure, and assess the extent of genetic diversity in 207 *Dioscorea rotundata* genotypes sourced from three different geographical origins. A total of 4,957 (16.2%) single nucleotide polymorphism markers were used to assess genetic diversity. The SNP markers were informative, with polymorphic information content ranging from 0.238 to 0.288 and a mean of 0.260 across all the genotypes. The observed and expected heterozygosity was 0.12 and 0.23, respectively while the minor allele frequency ranged from 0.093 to 0.124 with a mean of 0.109. The principal coordinate analysis, model-based structure and discriminant analysis of principal components, and the Euclidean distance matrix method grouped 207 yam genotypes into three main clusters. Genotypes from West Africa (Ghana and Nigeria) had significant similarities with those from Uganda. Analysis of molecular variance revealed that within-population variation across three different geographical origins accounted for 93% of the observed variation. This study, therefore, showed that yam improvement in Uganda is possible, and the outcome will constitute a foundation for the genetic improvement of yams in Uganda.

## Introduction

Yam (*Dioscorea* spp.) is a multi-species tuber-producing vine in the Dioscoreaceae family. The genus has over 603 domesticated and wild species found throughout the world's tropical and subtropical regions [1]. Yams are a food security crop that supports over 200 million people's lives in the tropics and subtropics, particularly in West Africa, where large-scale commercial

**Funding:** The authors (s) received no specific funding for this work.

**Competing interests:** The authors have declared that no competing interest exit

production is practiced [2]. Globally, annual yam output averages around 60 million tons, with a gross value of around 14 billion US dollars [3]. West Africa is the largest yam-growing region in the world accounting for 94% of global yam production [4] and the crop contributes significantly to the agricultural sector in terms of food, social, and cultural values [5]. Yam has the ability to grow in tropical and sub-temperate regions around the world suggesting that it is highly adaptable to its environment and that there may be adaptive traits (and associated alleles) that could be exploited in different global contexts [6]. Only ten of the over 600 *Dioscorea* species are farmed and economically significant [7]. According to Tamiru [8], *Dioscorea rotundata* is the most popular species in West and Central Africa, the main location for yam production worldwide.

Diversity at the genetic level is an essential requirement for a successful breeding programme [9]. Morphological and molecular characterization have been used as powerful tools in classifying cultivars and studying their taxonomic status [10]. Despite the importance and usefulness of yam, Asiedu et al [11] indicated that improper and limited characterized native genotypes at the morphological and molecular level had inhibited the breeding and selection of yam genotypes with improved traits. Similarly, Otoo [12] reported that the majority of traditional yam cultivars maintained by several ethnic groups in West Africa have not been adequately characterized, making it ambiguous, unreliable, and impossible to determine the existing genetic variation in yams.

Yam genetic diversity studies have mostly relied on morphological or low-throughput molecular markers. These included amplified fragment length polymorphism (AFLP) [13–17], random amplified polymorphic DNA (RAPD) [18,19], simple sequence repeats (SSRs) [7,15,20–23], inter simple sequence repeats (ISSRs) [21,24] and sequence-related amplified polymorphism (SRAP) [24]. They reported significant polymorphism and diversity among the yam genotypes used in their studies. In addition, Norman et al [25] characterized 52 accessions of *Dioscorea* spp. from Sierra Leone using 28 morphological descriptors and observed appreciable differences among the genotypes whilst Anokye et al [26] studied phenotypic diversity in a collection of yam genotypes from Ghana and Nigeria using morphological descriptors and grouped the accessions into two distinct clusters independent of geographic origin. Similarly, Veasey et al [27] examined isozymes to delineate yam cultivars and revealed that there was genetic diversity between and within yam species.

High-throughput genetic analysis has the potential to provide in-depth insight into population structure and genetic variability, and it has been used successfully in several crops. More recently, Agre et al [3] used high-density DArTseq SNP markers to characterize 100 *Dioscrorea alata* genotypes revealing significant genetic differences within the species. Also, Bhattacharjee et al [28] conducted a study using a genotyping-by-sequencing (GBS) approach to characterize 814 *Dioscorea rotundata* genotypes consisting of landraces, breeding lines, and commercial varieties. The aim was to understand the level of genetic diversity and pattern of the population structure among them, which elucidated that the majority of genetic variation existed within countries instead of between countries in the *Dioscorea rotundata* core collection and confirmed the reliability and accuracy of high-density SNP markers. Despite the extensive studies conducted on yam globally and in West African germplasm [25,26], there have been minimal research efforts in East African countries such as Kenya, Tanzania, DR Congo including Uganda. Efforts have been made to introduce improved yam cultivars, specifically *Dioscorea rotundata* clones, from West Africa for production in East Africa, especially Uganda [29]. These clones were distributed to farmers in different locations by the National Agriculture Research Organization (NARO, Uganda) and the International Institute of Tropical Agriculture (IITA), Uganda [30]. The National Root Crops Research Programme in Uganda is currently optimizing the yam breeding pipeline. To this effect, a considerable number of yam

genetic resources have been assembled, which include local landraces and improved genotypes from West Africa (Ghana and Nigeria). This genetic resource constitutes a founding population for the subsequent development of breeding populations and the generation of scientific information.

This study aimed to investigate the level of genetic diversity and population structure of diverse yam germplasm sourced from Uganda, Ghana, and Nigeria. To the best of our knowledge, this is the first report that used SNPs molecular variability in Ugandan yam germplasm to understand the genetic relationships among these genotypes in comparison to their geographical origin. The knowledge generated from this study will be extremely useful for future yam breeding programs, and the DArTseq SNPs markers will be useful in future genome-wide association studies to identify QTLs/ genes for desirable traits. The findings of this study will also help in making an informed choice of diverse parental lines to design new varieties in *Dioscorea rotundata* in Uganda.

## Materials and methods

### Genetic materials and leaf sampling

A total of 207 *Dioscorea rotundata* genotypes were used in this study. The genetic materials were part of the collection held at the National Crops Resource Research Institute (NaCRRI), which is an entity of the National Agricultural Research Organization (NARO) in Uganda (S1 Table). At NaCRRI, all genotypes were established using an augmented design with three local checks randomized in each block. The yam leaf sampling was carried out in accordance with the recommended plant sample collection kit (KBS-9370-001) protocol. Leaf samples from tagged plants were collected into 96-well tube plates 16 weeks after planting using a leaf puncher. After that, the leaves were oven-dried at a temperature of 80˚C.

### DNA extraction and SNP discovery by DArTseq™ technology

Yam-dried leaf samples were sent to SEQART AFRICA located at the International Livestock Research Institute (ILRI), Nairobi for genotyping. DNA extraction was done using Nucleomag Plant DNA extraction kit (Mag-Bind® Plant DNA DS 96 Kit). The genomic DNA extracted was in the range of 50-100ng/ul. DNA quality and quantity were checked on 0.8% agarose gel. Libraries were constructed following DArTSeq complexity reduction method [31] through digestion of genomic DNA using a combination of *Pst*I and *Mse*I enzymes. This was followed by ligation of barcoded adapters and common adapters followed by PCR amplification of adapter-ligated fragments. Libraries were then sequenced using single-read sequencing runs for 77 bases.

Next-generation sequencing was carried out using Hiseq2500 [31]. SEQART AFRICA platform uses genotyping by sequencing (GBS) DArTseq™ technology, which provides rapid, high quality, and affordable genome profiling, even from the most complex polyploid genomes. DArTseq markers scoring was achieved using DArTsoft14, which is an in-house marker scoring pipeline [31]. Two types of DArTseq markers were scored, SilicoDArT markers and SNP markers, both scored as binary for presence/absence (1 or 0, respectively) of the restriction fragment with the marker sequence in the genomic representation of the sample. Both SilicoDArT markers and SNP markers were aligned to the *Dioscorea rotundata* reference genome (TDr96_F1_v2_PseudoChromosome.rev07) to identify chromosome positions.

### Filtering, analysis of genetic diversity and population structure

Data quality control and filtering were performed using TASSEL (v5.2.52) [32]. Single nucleotide polymorphism (SNP) markers with >20% of missing data, <0.05 minor allele frequency

(MAF), and unknown positions on the genome were removed. SNP data were further imputed using the k-nearest neighbor genotype imputation method [32]. Eventually, a total of 4,957 SNPs were selected for further analysis.

SNP marker information and diversity analysis were carried out with parameters such as minor allele frequencies using TASSEL (v5.2.52) [32] and the polymorphism information content (PIC) was assessed using PowerMarker (v3.25) [33]. Also observed and expected heterozygosity was determined using the "*adegenet*" package in R [34]. Call rate and reproducibility were also calculated in package "*dartR*" in the R package [35]. Genetic structure was assessed using the Bayesian model-based clustering implemented in the STRUCTURE software (v2.3.4) [36,37]. The number of hypothetical sub-populations (K) was estimated through the application of a Bayesian clustering approach. The STRUCTURE analysis was run considering a burn-in period of 10,000 Markov-chain Monte Carlo iterations and a 10,000-run length with an admixture model following the Hardy–Weinberg equilibrium and its correlated allele frequencies. Ten independent runs were performed for the value of each number of clusters, which ranged from 1 to 10. The structure outputs were analyzed using STRUCTURE HARVESTER [38], which enabled the identification of the best K value as the distinct peak in the change of likelihood (ΔK). Moreover, individuals with $q_i \geq 0.6$ were assigned to a specific genetic group.

The STRUCTURE analysis was complemented with a discriminant analysis of principal components (DAPC) using the "*adegenet*" package in R [34] with the determination of the optimal number of clusters inferred using the K-means analysis by varying the possible number of clusters from one to 20. DAPC scatter plots were later developed on the clusters identified through K-means using the first 20 principal components. The results of the STRUCTURE analysis and DAPC analysis were compared. Analysis of Molecular Variance (AMOVA) [39] was conducted to assess the population differentiation among the genetic groups based on the geographical origins of the 207 yam genotypes using GenAlEx 6.503 [40]. Before AMOVA, the marker datasets were numerically coded as A = 1, C = 2, G = 3, and T = 4 [41]. Numerically coded data were subjected to AMOVA with 999 permutations. The genetic variations were partitioned into two: variation among the population (PhiPR) and variation within the population (PhiPT) [42]. Also, phylogenetic relationships among genotypes based on geographical origins were built using a Euclidean distance matrix in PowerMarker (v3.25) [33]. The resulting tree was visualized using Molecular Evolutionary Genetics Analysis (MEGA-X v10.1.8) software [43].

## Results

### Quality, diversity and functional characterization of DArTseq-SNPs on yam chromosome

A total of 30,542 unfiltered SNPs were generated from the DArTseq genotyping of 207 yam genotypes. A large set of SNPs were removed during filtering, and 4,957 good-quality SNPs (16.2%) distributed across the 20 chromosomes of *Dioscorea rotundata* (Fig 1 and S2 Table), were selected for further analyses. Marker density across chromosomes revealed the highest number of SNPs in chromosome 5 (524 SNPs; 10.6%) followed by chromosome 19 (485 SNPs; 9.8%) and chromosome 4 (367 SNPs; 7.4%) whilst the lowest number of SNPs were mapped in chromosome 13 (91 SNPs; 1.8%) (Fig 1 and S2 Table). The chromosome length varied from 8.3 Mbps (chromosome 13) to 32.9 Mbps (chromosome 20). Quality parameters such as average PIC value across all the markers was 0.259 and ranged from 0.238 to 0.288, while the call rate and average reproducibility across all the markers were 0.702% and 0.991%, respectively. The observed heterozygosity ranged from 0.097 to 0.153 with a mean of 0.115 whilst the expected heterozygosity ranged from 0.256 to 0.314 and the mean was 0.280. Similarly, the minor allele frequency (MAF) ranged from 0.093 to 0.118 with an average of 0.109 (S2 Table).

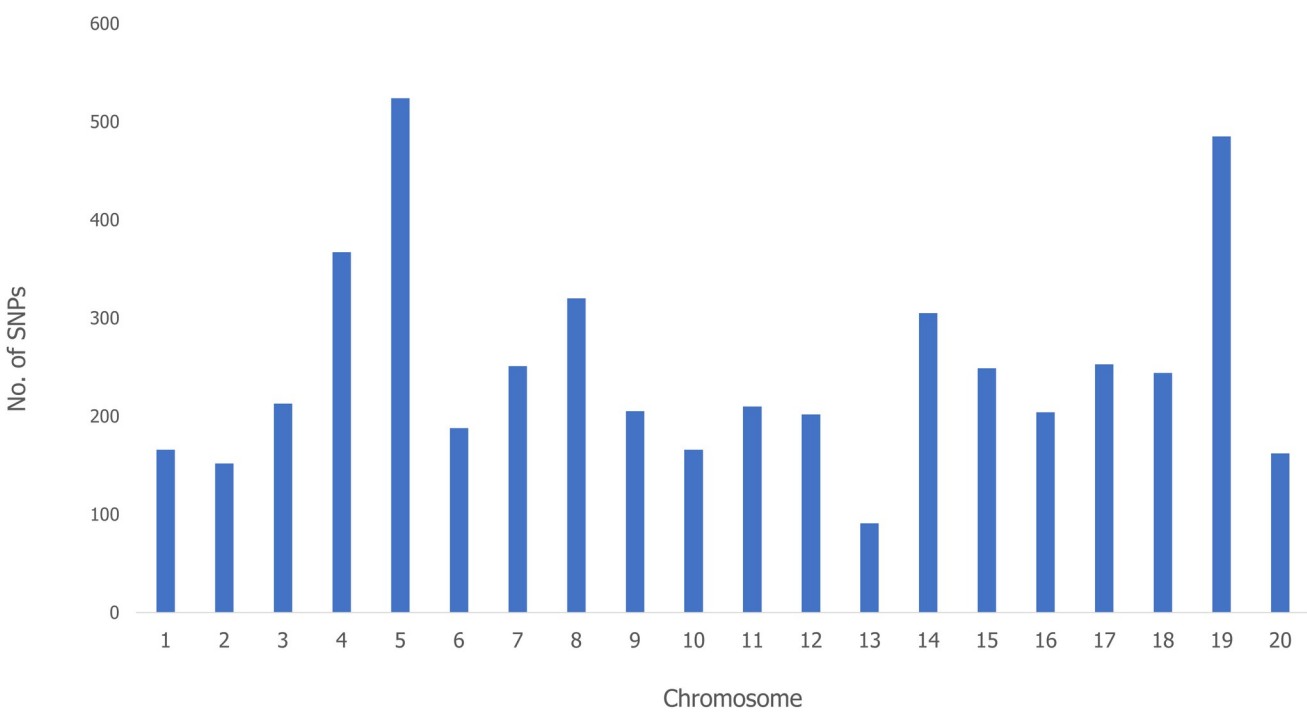

**Fig 1. Distribution of 4,957 SNPs across 20 *Dioscorea rotundata* chromosomes.**

## Population differentiation and genetic relation between yam genotypes

The distribution of heterozygosity across 207 yam genotypes and 4,975 SNP markers revealed low heterozygosity of 0.2 and 0.4, respectively (Fig 2). The kinship coefficient between pairs of yam genotypes varied from -0.025 to 0.758. Overall, 58% of the pairs of 207 yam genotypes had kinship values of 0.6 (S1 Text). The kinship matrix obtained from DArTseq-SNP markers resulted in two distinct groups (Fig 3). In order to identify the most similar pairs of genotypes, a genetic distance matrix was computed between pairs of genotypes, which varied from 0.08 to 0.93, with an average of 0.54 (S2 Text). A large proportion (75%) of pairs of genotypes showed a genetic distance of 0.45 (S2 Text).

The analysis of molecular variance (AMOVA) among three populations/genetic groups based on geographic origin indicated that 93% of the variability was within each genetic group and only 7% was between them., and this was statistically significant (P < 0.001) (Table 1).

The cluster analysis based on the Euclidean distance matrix differentiated the 207 yam genotypes into two major clusters (clusters 1 and 2) (Fig 4) with sub-clusters within the main clusters. Cluster one was the largest cluster consisting of a mixture of genotypes sourced from all the geographical origins (Ghana, Nigeria, and Uganda). Cluster two consisted of only seven genotypes and were all from West Africa. There was no relationship between the cluster grouping and the origin of the yam genotypes.

## Population structure, principal coordinate analysis and discriminate analysis of principal coordinate

To understand the pattern of population structure, a Bayesian Information Criterion (BIC) and complementary coordination analysis by discriminant analysis of principal components (DAPC) was performed. The determination of the populations at each K-value and

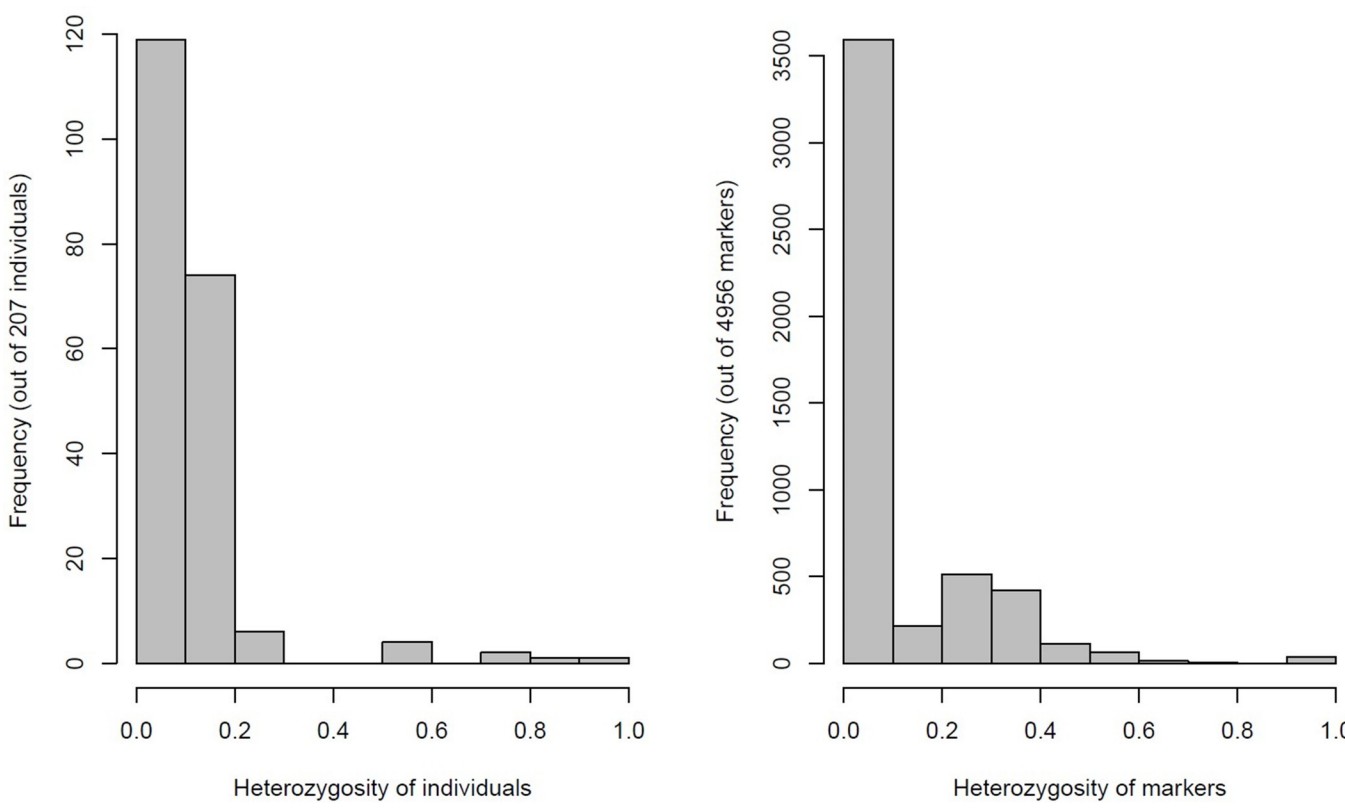

**Fig 2. Frequency of heterozygous genotypes and heterozygosity of 4,957 SNP markers generated using DArTseq platform across 207 yam genotypes.**

membership coefficients (qi) in STRUCTURE analysis was very informative. Simulations (logarithm probability relative to standard deviation, ΔK) estimated from the SNP markers showed a sharp peak at ΔK = 2 (Fig 5A), which explained the optimum number of sub-populations (ΔK = 2). At ΔK = 2, POP I and POP II consisted of 195 genotypes (94.0%) and 12 genotypes (6.0%), respectively. The highest number of genotypes observed in POP I was sourced from Ghana (81 genotypes; 41.5%) followed by Nigeria (76 genotypes; 40%) and the least was from Uganda (38 genotypes; 19.5%) (Fig 5B, S3 Table). In POP II, the observed number of genotypes was equally distributed among the various sources of the genotypes. At ΔK = 3, it was observed that all the genotypes in POP II were maintained (12 genotypes; 6% similar to POP II at ΔK = 2) whilst POP I at ΔK = 2 further subclustered into POP I (61 genotypes; 27.8%) and POP III (134 genotypes; 66.2%), respectively. In POP I at ΔK = 3, the highest number of genotypes (28 genotypes; 45.9%) were from Uganda and the least from Nigeria (14 genotypes; 23.0%). Only 19 genotypes (31.1%) were from Ghana (Fig 5B, S3 Table). Similar observations were made at ΔK = 4 where a total of 60 genotypes, 127 genotypes, 12 genotypes, and 8 genotypes were observed from POP I, II, III, and IV, respectively. The expected heterozygosity was 0.12 and 0.15 for POPI and POPII, respectively, with high Fst values of 0.78 and 0.75 (Table 2).

The DAPC method was further carried out to assess the sub-clusters at K = 2 (Fig 6A), K = 3 (Fig 6B), and K = 4 (Fig 6C). The summary of the DAPC cluster groupings and probabilities of cluster membership allocations of genotypes at K = 2, 3, and 4 are presented in the S3 Table. Based on the possibility of cluster membership assignment, DAPC cluster grouping both K = 2 and K = 3 represented a good fit. At K = 2, cluster one consisted of 196 genotypes (94.0%) including 38 genotypes (19.4%) sourced from Uganda and majority representing

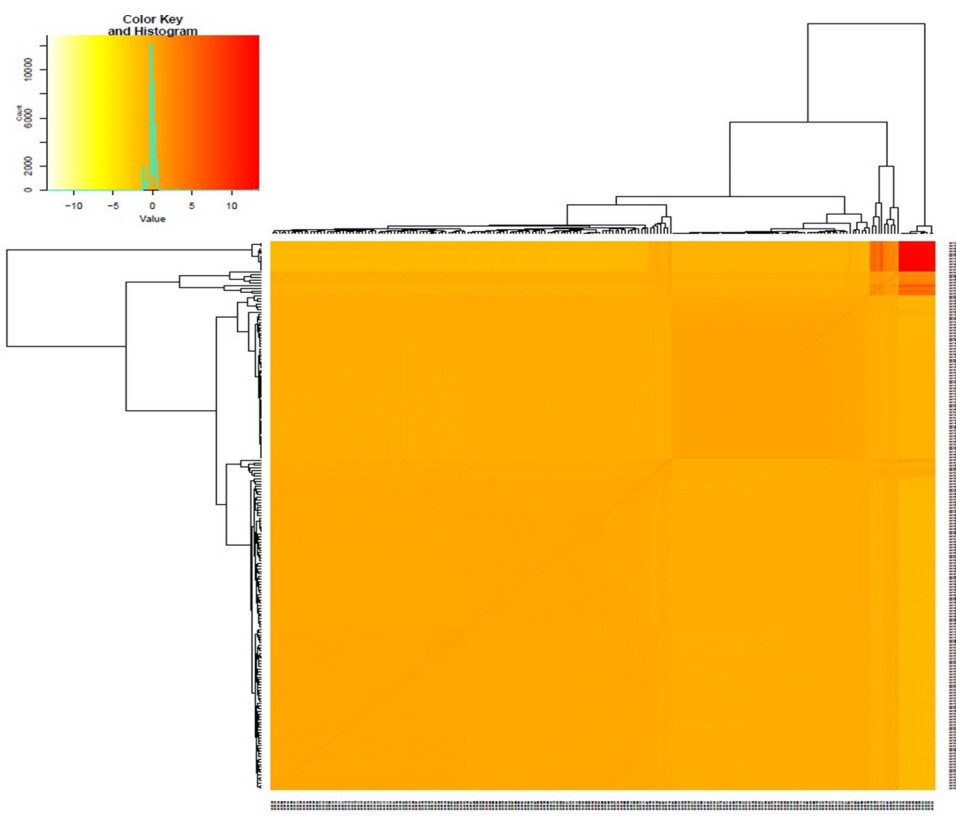

**Fig 3. Heat map plot of kinship matrix using average linkage clustering based on SNP markers.**

genotypes from Ghana (82 genotypes; 41.8%) followed by Nigeria (76 genotypes; 38.8%) (Fig 6B). Cluster two only consisted of 11 genotypes (6.0%). These were in accordance with results from STRUCTURE analysis. The DAPC biplot, together with the plot of densities of individuals on the first discriminant function, showed a clear separation of the yam genotypes into three clusters with minimal admixed individuals at K = 3. Furthermore, the three discriminant functions, which explained cluster one (59 genotypes; 28.5%), cluster two (137 genotypes; 66.2%), and cluster three (11 genotypes; 5.3%) obtained (Fig 6C). Cluster two which contributed the highest number of genotypes consisted of primarily *Dioscorea rotundata*

**Table 1. Analysis of molecular variance within and among the 207 genotypes assessed with 4,957 SNP markers.**

| SOV[a] | df[b] | SS[c] | MS[d] | Est. Var.[e] | %[f] |
|---|---|---|---|---|---|
| **Among Pops[g]** | 2 | 8033.172 | 4016.586 | 50.721 | 7% |
| **Within Pops** | 204 | 133070.364 | 652.306 | 652.306 | 93% |
| **Total** | 206 | 141103.536 | | 703.027 | 100% |

[a]Sources of variance

[b]Degree of freedom

[c]Sum of squares

[d]Means squares

[e]Estimated variance

[f]estimated variance

[g]Population.

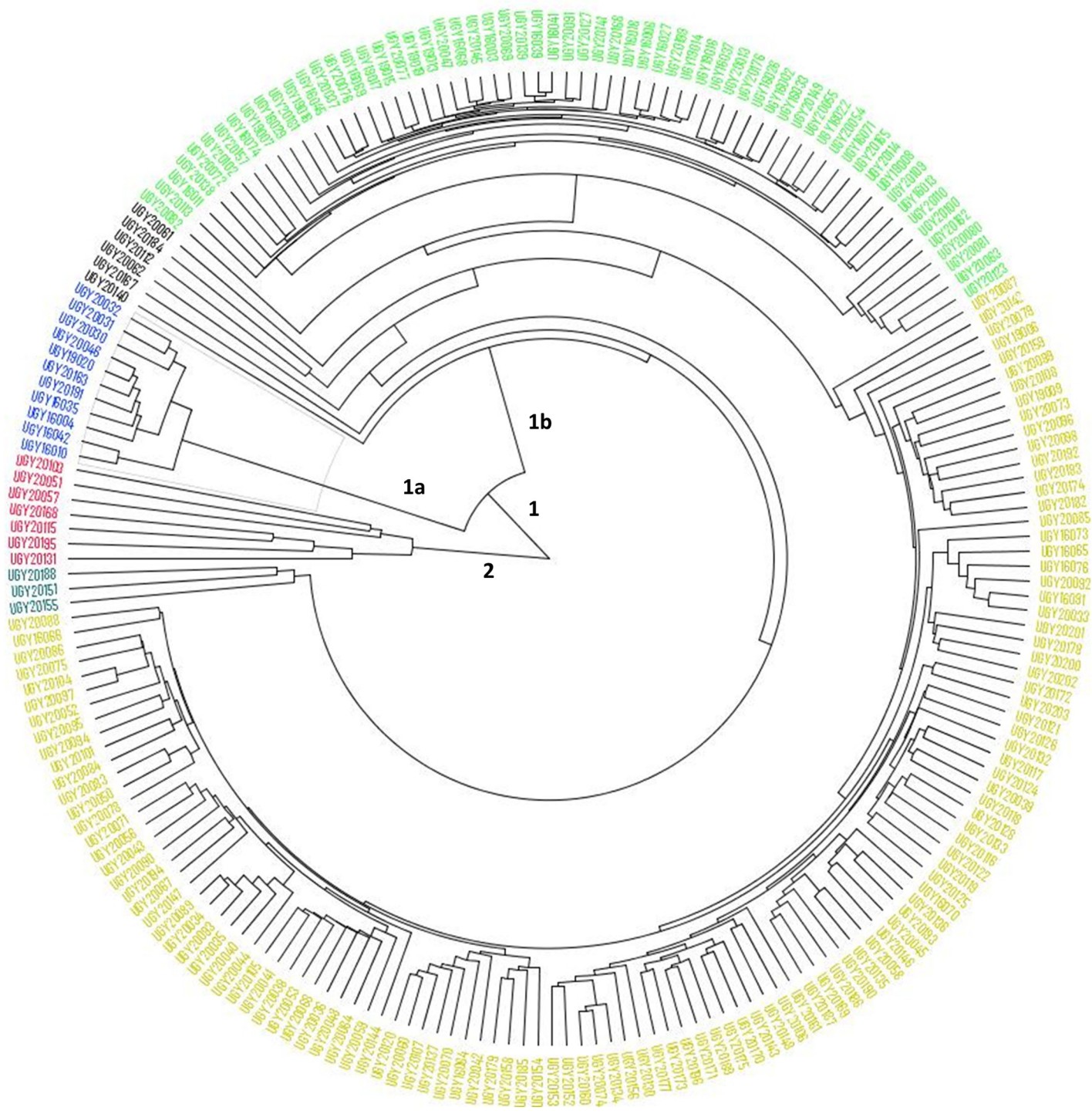

**Fig 4. Hierarchical clustering of 207 yam genotypes based on 4,957 SNPs.**

sourced from West Africa (Ghana and Nigeria) with only 10 genotypes (7.3%) sourced from Uganda. However, the cluster groupings and probabilities of cluster membership allocations of genotypes in K = 3 were maintained and equally represented in K = 4 (Fig 6D). The PIC proportion among the clusters ranged from 0.16 to 0.31, with a mean of 0.24 (Table 3). Cluster

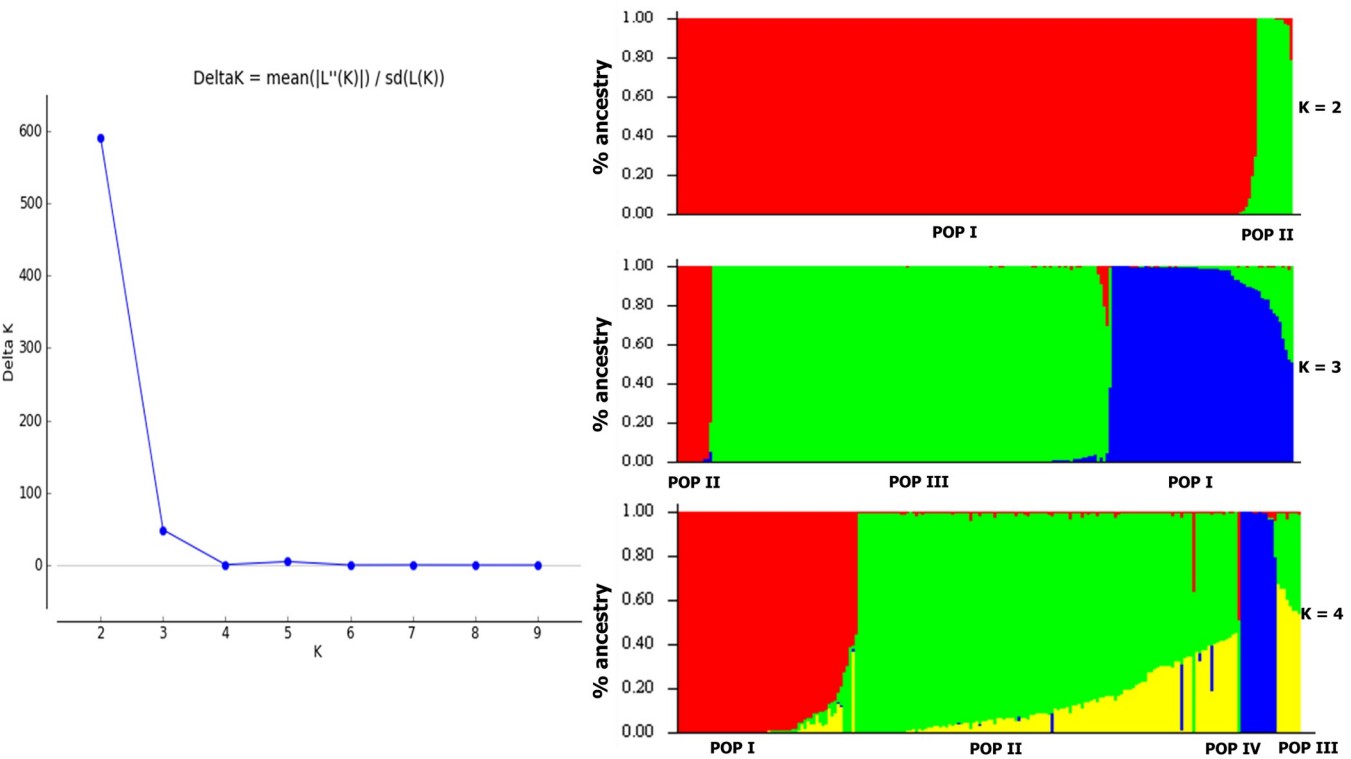

**Fig 5. Population structure in the yam germplasm.** (A) Likelihood of ΔK showing the best K value (K = 2); (B) Population structure obtained for 207 yam genotypes based on 4,957 SNPs.

two represented the highest PIC value (0.31), and the least was cluster three (0.16). The genetic diversity among the clusters followed the same trend as PIC with the highest genetic diversity for cluster two (0.38), followed by cluster one (0.26), and the least was cluster three (0.18) with a mean of 0.27. A significant net nucleotide distance between clusters was observed and it was high between clusters one and two (0.53), followed by the distance between clusters two and three (0.34), and the least distance between clusters one and three (0.05) (Table 3).

A principal coordinate analysis (PCoA) based on the pairwise Euclidean genetic distance matrix among the genotypes was performed to depict the genetic divergence in the yam genotypes using the DArTseq-SNP markers. The PCoA results revealed 88.2% of the total genetic variation, which was explained by the first two axes of the PCoA (S1 Fig). The first axes explained 83.2% (eigenvalue = $1.20e^8$) of the genetic variation whilst the second axes explained 4.8% (eigenvalue = $6.89e^6$) of the total genetic variation. The distribution of the genotypes

**Table 2. Pairwise comparison (Fst) and expected heterozygosity among the two subpopulations using the model-based clusters.**

|        | POP[a]I | Expected He[b] | Fst[c] | %[d] Membership |
|--------|---------|----------------|--------|-----------------|
| POPI   | -       | 0.12           | 0.78   | 0.94            |
| POPII  | 0.56    | 0.15           | 0.75   | 0.06            |

[a]Population

[b]heterozygosity

[c]Fixation index

[d]Percentage.

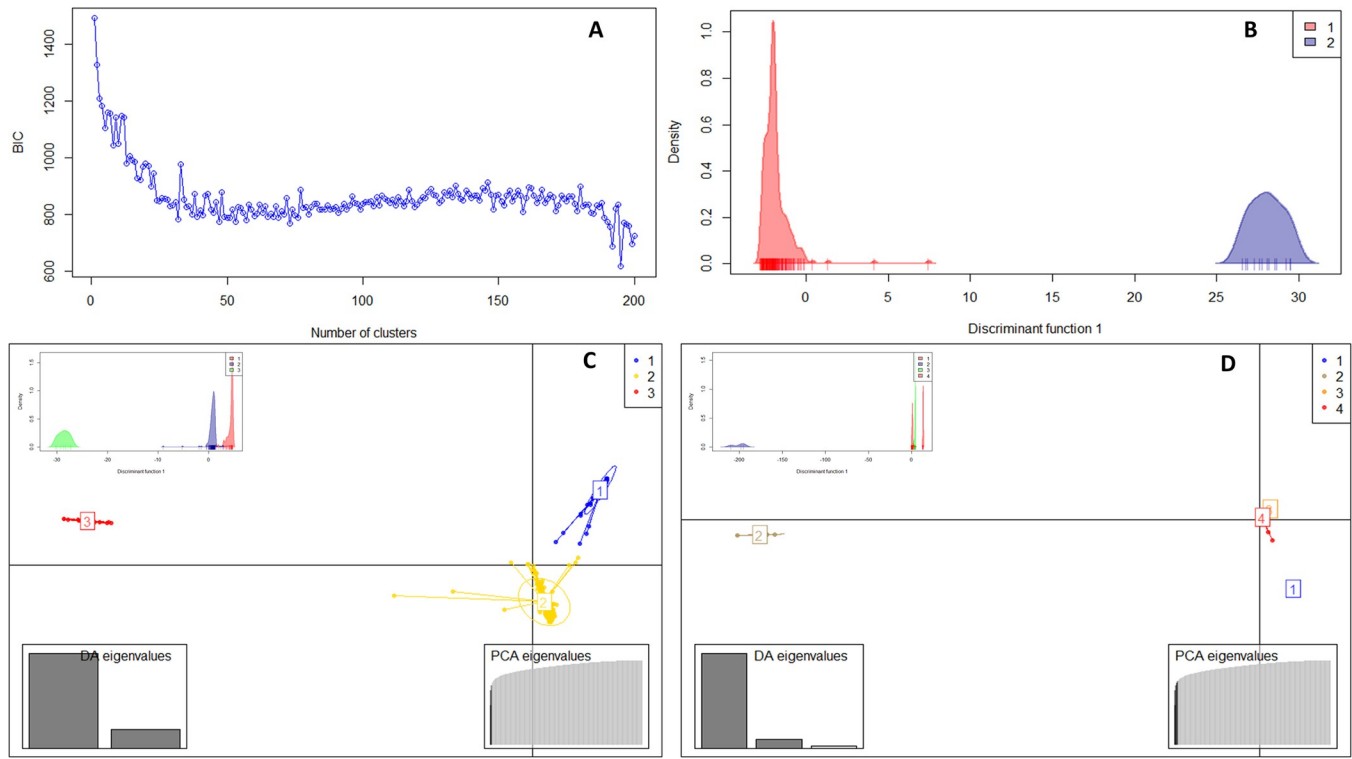

**Fig 6. Discriminant analysis of principal components (DAPC).** **(A)** The graph of Bayesian information criterion (BIC) vs. the number of clusters indicates the optimum number of clusters inferred in the yam diversity panel; **(B)** Discriminant analysis of principal components with K = 2; **(C)** Discriminant analysis of principal components with K = 3; **(D)** Discriminant analysis of principal components with K = 4. The axes represent the first two linear discriminants. Each colour represents a cluster and each does represent an individual.

based on SNP markers and principal component analysis showed minimal diversity among the genotypes based on their geographical origins. The graph revealed three major clusters indicating that the genotypes sourced from Nigeria (blue) and genotypes sourced from Ghana (red) grouped together, while the genotypes from Uganda (green) were scattered among the genotypes from Nigeria and Ghana across different quadrants (S1 Fig).

## Discussion

Characterization of genetic diversity in crop species is critical for efficient germplasm conservation and further use in breeding programs [44]. Genetic linkages between several *Dioscorea*

**Table 3. Genetic variability among (net nucleotide distance) and proportion of individuals, Major allele frequency, gene diversity, and PIC.**

| Marker | Net nucleotide distance | | Individuals (%[a]) | GD[b] | PIC[c] |
|---|---|---|---|---|---|
| | Cluster 2 | Cluster 3 | | | |
| **Cluster 1** | 0.53 | 0.05 | 17.0 | 0.25 | 0.23 |
| **Cluster 2** | - | 0.34 | 72.1 | 0.39 | 0.31 |
| **Cluster 3** | - | - | 10.9 | 0.18 | 0.16 |
| **Mean** | | | | 0.28 | 0.24 |

[a]Percentage

[b]Gene diversity

[c]Polymorphic information content.

*rotundata* yam genotypes derived from three geographical locations were analyzed in this study to develop a yam breeding program in Uganda. The 4,957 SNP markers identified in the current study were distributed across the 20 chromosomes as observed in Agre et al [3], Tamiru et al [45] and Bhattacharjee et al [28].

In the present study, a high-throughput genotyping using DArTseq SNP markers was used to understand the genetic relationships among yam genotypes sourced from different geographical origins. The 4,957 high-quality selected SNPs were polymorphic as observed in previous SNP-based genetic diversity studies in *Dioscorea alata* [3] and *Dioscorea rotundata* [28]. Similar polymorphism values have also been reported in several important food crops such as oilseed crop (0.21) [46], soybean (0.28) [47], rice (0.31) [48], maize (0.19) [49], cowpea (0.24) [50], common bean (0.25) [51] and chickpea (0.32) [52]. This study also demonstrated the possibility of using the selected DArTseq-SNP markers for genomic investigations in yams, which may serve as a foundation for future breeding efforts in Uganda and conservation initiatives in the country. The study elucidated that the majority of the genetic variances exist within countries instead of between countries in *Dioscorea rotundata* genotypes sourced from Ghana, Nigeria, and Uganda. Significantly, low heterozygosity (He) observed in this study suggests that the proportion of heterozygous genotypes sourced from different geographical origins was relatively low. This is slightly different from the He values reported for similar studies in *Dioscorea rotundata* [28] and on *Dioscorea alata* [3]. This may be related with the diverse set of genotypes used in these studies. The low heterozygosity observed in this study could also be attributed to the fact that majority of the yam genotypes may have been commonly shared among all the geographical origins such as Ghana and Nigeria, and the transfer of yam genotypes from IITA to Uganda. For yam genotypes from Uganda, the genetic base may be narrow because only few selected genotypes may be cultivated by local natives, resulting in duplications/mislabeling/mixtures.

The analysis of population structure based on SNPs provides helpful information in maintaining and monitoring the genetic diversity required for a robust breeding program [53]. The selected SNP markers in this study were informative, suggesting that they were suitable for assessing the genetic variation among and within the yam genotypes used in this study. Two approaches (STRUCTURE and DAPC) were used to determine the population structure, in which the STRUCTURE analysis revealed the presence of two major populations (K = 2) for the 207 yam genotypes. Similar results were observed in cassava [54,55] but not in *Dioscorea alata* (yam) [3], cowpea [50] and chickpea [52] wherein ΔK was 3, 3, and 4, respectively. The observed genetic differentiation was further confirmed by the high fixation index (Fst) value among genotypes between clusters. This is in contrast with studies by Agre et al [3] in *Dioscorea alata*, Fatokun et al. [56] in cowpea and Rabbi et al. [57] in cassava, wherein low fixation index was reported. The low expected heterozygosity observed in the current study could be due to the complex breeding history of the genotypes in Nigeria and Ghana. Although the genetic diversity observed in this study was low, but it has provided valuable information that can be used in breeding and research activities in Uganda. The germplasm can be evaluated for specific traits and guide in the selection of diverse genotypes to constitute a training population for genomic prediction and genome-wide association studies.

Based on DAPC, three clusters were obtained showing a moderate genetic differentiation between clusters. Therefore, genetic variation was majorly captured within sub-populations. Similarly, population genetic analysis based on PCoA revealed three major clusters with considerable admixtures among the genotypes. This was similar to the results obtained by Agre et al [3] which reported K = 3 clusters. The genetic variation in this study was also explained using AMOVA analysis that partitioned total variation within the population rather than among population indicating close evolutionary relationships and admixture among yam

genotypes, especially those that were sourced from Ghana and Nigeria. Similar results were reported by Kedra et al [58] in yams from Ethiopia, Loko et al [59] in guinea yams from Benin, Muluneh et al [60] in yams from Ethiopia, and Wendawek et al [61] in cultivated and wild yams in Ethiopia wherein within-population variation were higher than among population.

This was the first study that used high-throughput genotyping approach to assess the genetic diversity in yam genotypes from Uganda. This will enable to identify diverse and unique genetic materials for use as parents in future breeding process. However, there is a need for a broader study within East Africa including countries such as Uganda, Kenya, Tanzania, and DR Congo using a wide range of genotypes. This will also provide the possibility of developing a regional strategy for conservation of yam germplasm and sharing genotypes with comparable genetic traits across countries/breeding programs for further development and release to farmers.

## Supporting information

**S1 Fig. Principal coordinate analysis (PCoA) of Yam genotypes based on DArT SNP markers.** (Uganda = Green, Nigeria = blue, Ghana = Red).
(TIF)

**S1 Table. List of genotypes and their geographical origins.**
(DOCX)

**S2 Table. Quality and summary statistics of DArTseq-SNPs on *Dioscorea rotundata* chromosomes.**
(DOCX)

**S3 Table. List of genotypes, the proportion of grouping on based STRUCTURE analysis and geographical origin.**
(DOCX)

**S1 Text. Kinship matrix obtained from DArTseq-SNP markers for yam genotypes.**
(TXT)

**S2 Text. The genetic distance matrix computed between pairs of yam genotypes.**
(TXT)

## Acknowledgments

We acknowledge the support and resources from Scientists in Crop Improvement for Food Security in Africa (SCIFSA) for the Ph. D fellowship, National Crops Resource Research Institute (NARO-NaCRRI), Makerere Regional Center for Crop Improvement (MaRCCI), Council for Scientific and Industrial Research–Savanna Agricultural Research Institute (CSIR-SARI), and International Institute of Tropical Agriculture IITA/Ibadan/Sendusu) for their advice and guidance in this study. We also acknowledge the contributions of Dr. Ophelia Osei Ulzen (CSIR-SRI), Dr. Frejus Ariel Kpedetin Sodedji, and Baba Yussif (CSIR-SARI) for their helpful comments on the data analysis

## Author Contributions

**Conceptualization:** Emmanuel Amponsah Adjei, Williams Esuma, Ranjana Bhattacharjee, Thomas Lapaka Odong.

**Data curation:** Emmanuel Amponsah Adjei.

**Formal analysis:** Emmanuel Amponsah Adjei, Isaac Onziga Dramadri.

**Investigation:** Emmanuel Amponsah Adjei.

**Methodology:** Emmanuel Amponsah Adjei, Williams Esuma, Ranjana Bhattacharjee, Thomas Lapaka Odong.

**Resources:** Emmanuel Amponsah Adjei, Titus Alicai, Isaac Onziga Dramadri, Richard Edema, Emmanuel Boache Chamba, Thomas Lapaka Odong.

**Software:** Emmanuel Amponsah Adjei, Isaac Onziga Dramadri.

**Supervision:** Williams Esuma, Thomas Lapaka Odong.

**Validation:** Emmanuel Amponsah Adjei, Williams Esuma, Isaac Onziga Dramadri, Thomas Lapaka Odong.

**Visualization:** Emmanuel Amponsah Adjei, Thomas Lapaka Odong.

**Writing – original draft:** Emmanuel Amponsah Adjei.

**Writing – review & editing:** Emmanuel Amponsah Adjei, Williams Esuma, Titus Alicai, Ranjana Bhattacharjee, Isaac Onziga Dramadri, Richard Edema, Emmanuel Boache Chamba, Thomas Lapaka Odong.

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
