## [Decision Letter · Decision Letter 0]

14 Sep 2022

PONE-D-22-02542Genetic diversity and population structure of Uganda’s yam (Dioscorea spp.) genetic resource based on DArTseqPLOS ONE

Dear Dr. Emmanuel,

Thank you for submitting your manuscript to PLOS ONE. After careful consideration, we feel that it has merit but does not fully meet PLOS ONE’s publication criteria as it currently stands. Therefore, we invite you to submit a revised version of the manuscript that addresses the points raised during the review process.

We look forward to receiving your revised manuscript.

Kind regards,

Faheem Shehzad Baloch, PhD

Academic Editor

PLOS ONE

Journal Requirements:

https://journals.plos.org/plosone/s/file?id=ba62/PLOSOne_formatting_sample
**
_title_authors_affiliations.pdf
**

Reviewers' comments:

Reviewer's Responses to Questions

**Comments to the Author**

1. Is the manuscript technically sound, and do the data support the conclusions?

Reviewer #1: Yes

Reviewer #2: Yes

2. Has the statistical analysis been performed appropriately and rigorously? 

Reviewer #1: Yes

Reviewer #2: Yes

3. Have the authors made all data underlying the findings in their manuscript fully available?

Reviewer #1: No

Reviewer #2: Yes

4. Is the manuscript presented in an intelligible fashion and written in standard English?

Reviewer #1: No

Reviewer #2: Yes

5. Review Comments to the Author

Reviewer #1: 1- Cite this paper: Bredeson, J.V., Lyons, J.B., Oniyinde, I.O. et al. Chromosome evolution and the genetic basis of agronomically important traits in greater yam. Nat Commun 13, 2001 (2022).

2- Line 69-75, is too long sentence. Just present the results in short sentences, which is easy to read and understanding. Please re-write this!!

3- In the Materials & methods, DArTseq marker genotyping explained, but it seems that this part done by Diversity Array Technology Inc, Canberra, Australia. If so, it should be clearly noted and cited in Materials & Methods.

See these example papers of different plant species:

- Hassani, S.M.R., Talebi, R., Pourdad, S.S. et al. In-depth genome diversity, population structure and linkage disequilibrium analysis of worldwide diverse safflower (Carthamus tinctorius L.) accessions using NGS data generated by DArTseq technology. Mol Biol Rep 47, 2123–2135 (2020).

- Mahboubi, M., Mehrabi, R., Naji, A.M. et al. Whole-genome diversity, population structure and linkage disequilibrium analysis of globally diverse wheat genotypes using genotyping-by-sequencing DArTseq platform. 3 Biotech 10, 48 (2020).

4- Also its recommended to cite some research paper for the application of DArTseq markers on different plant species and then cite the paper of Agre et al. (2019).

5- In Table 1, it’s better to add some more information about the markers coverage on each chromosomes and also the amount of mean and range of PIC for each chromosome (see example paper : (Farahani et al. 2019; Hassani et al. 2020).

6- Line 244-250, should be removed to Introduction part.

7- Line 278-294: This is just the repetition of results. Here authors should more focus on the application of these generated data for any future germplasm conservation and breeding programs in Yam. For example, there were no information about the certain genotypes that are very diverse as a hetrotic groups that can used in breeding programs or mentioned genotypes/sources from very close sources, if their morphological characteristics (if available) are the same or difference!!!. These part should be more discussed.

8- An table for markers information like name, chromosome, physical position, PIC and some quality parameters like: call rate and reproducibility should be presented as supplementary Table

9- Text need to extensive English grammar check.

Reviewer #2: Comments for authors

The submitted manuscript investigated population structure, nature, and extent of genetic diversity in 207 Dioscorea rotundata genotypes sourced from three different geographical origins using diversity array technology (DArTSeq).

The overall framework of the study was well planned, the number of materials used is enough, and assessed genetic diversity among different genotypes using three complementary clustering methods, STRUCTURE analysis, discriminant analysis of principal components (DAPC), and cluster analysis.Here are some specific comments with the hope that will help to clarify some things and improve the manuscript overall.

The authors did not express their aim well enough in the last paragraph of the introduction section and should be written more clearly about how the results of the study will contribute to the literature.

Should be added to more literature related to this study in the introduction section.

Please revise some Latin words as italic in the text.

L92.P4.The authors stated that “genotypes sourced from three different geographical origin” in the abstract file. Please write these origins in detail by specifying the genotype number.

L94.P4. The S1 provided the genotypes ID, description, and geographical origin. Please give more detail about passport data of plant materials; Collection Site, District, Village, Altitude (m), and Coordinates

L98.P5. Please give more detail about the growing condition before the DNA extraction.

L98.P5. Please give more detail about the DNA extraction protocol.

L103.P8. Please give more detailed information about how minor allele frequency, observed heterozygosity, expected heterozygosity, and gene diversity were determined in the statistical analysis section.

Please provide all figures with 300dpi resolution.

L233. P11. Please revise as “..mixture of all genotype sources from..” instead of “mixture of all genotype’s sources from”

L246-250.P12. Please also provide these literatures in the introduction section.

L262. P12. To support your statements add a more recent study for common bean; Nadeem et al. (2018) https://doi.org/10.1371/journal.pone.0205363

The discussion section should be improved.

The conclusion should be shortened and concrete.

The reference section should be double-checked for the journal formatting rules

6. PLOS authors have the option to publish the peer review history of their article (what does this mean?). If published, this will include your full peer review and any attached files.

Reviewer #1: No

Reviewer #2: **Yes: **Mehmet Zahit Yeken

---

## [Decision Letter · Decision Letter 1]

31 Oct 2022

Genetic diversity and population structure of Uganda’s yam (Dioscorea spp.) genetic resource based on DArTseq

PONE-D-22-02542R1

Dear Dr. Amponsah Adjei,

We’re pleased to inform you that your manuscript has been judged scientifically suitable for publication and will be formally accepted for publication once it meets all outstanding technical requirements.

Kind regards,

Mehdi Rahimi, Ph.D.

Academic Editor

PLOS ONE

Additional Editor Comments (optional):

Reviewers' comments:

Reviewer's Responses to Questions

**Comments to the Author**

1. If the authors have adequately addressed your comments raised in a previous round of review and you feel that this manuscript is now acceptable for publication, you may indicate that here to bypass the “Comments to the Author” section, enter your conflict of interest statement in the “Confidential to Editor” section, and submit your "Accept" recommendation.

Reviewer #1: All comments have been addressed

Reviewer #2: All comments have been addressed

2. Is the manuscript technically sound, and do the data support the conclusions?

Reviewer #1: Yes

Reviewer #2: Yes

3. Has the statistical analysis been performed appropriately and rigorously? 

Reviewer #1: (No Response)

Reviewer #2: Yes

4. Have the authors made all data underlying the findings in their manuscript fully available?

Reviewer #1: Yes

Reviewer #2: Yes

5. Is the manuscript presented in an intelligible fashion and written in standard English?

Reviewer #1: Yes

Reviewer #2: Yes

6. Review Comments to the Author

Reviewer #1: (No Response)

Reviewer #2: (No Response)

7. PLOS authors have the option to publish the peer review history of their article (what does this mean?). If published, this will include your full peer review and any attached files.

Reviewer #1: No

Reviewer #2: **Yes: **Mehmet Zahit Yeken

---

## [Editor Report · Acceptance letter]

5 Feb 2023

PONE-D-22-02542R1 

Genetic diversity and population structure of Uganda’s yam (*Dioscorea spp.*) genetic resource based on DArTseq 

Dear Dr. Amponsah Adjei:

I'm pleased to inform you that your manuscript has been deemed suitable for publication in PLOS ONE. Congratulations! Your manuscript is now with our production department. 

Kind regards, 

on behalf of

Associate Prof. Mehdi Rahimi 

Academic Editor

PLOS ONE